# Effect of Nut Consumption on Erectile and Sexual Function in Healthy Males: A Secondary Outcome Analysis of the FERTINUTS Randomized Controlled Trial

**DOI:** 10.3390/nu11061372

**Published:** 2019-06-19

**Authors:** Albert Salas-Huetos, Jananee Muralidharan, Serena Galiè, Jordi Salas-Salvadó, Mònica Bulló

**Affiliations:** 1Human Nutrition Unit, Biochemistry and Biotechnology Department, Faculty of Medicine and Health Sciences, Universitat Rovira i Virgili (URV), 43201 Reus, Spain; albert.salas@utah.edu (A.S.-H.); jananee.muralidharan@estudiants.urv.cat (J.M.); serena.galie@outlook.it (S.G.); 2Institut d’Investigació Sanitària Pere i Virgili (IISPV), 43204 Reus, Spain; 3Consorcio CIBER, M.P., Physiopathology of Obesity and Nutrition (CIBERobn), Instituto de Salud Carlos III (ISCIII), 28029 Madrid, Spain; 4Hospital Universitari Sant Joan de Reus (HUSJR), 43204 Reus, Spain

**Keywords:** nuts, RCT, erectile function, sexual desire, orgasmic function, nitric oxide, E-selectin

## Abstract

Lifestyle risk factors for erectile and sexual function include smoking, excessive alcohol consumption, lack of physical activity, psychological stress, and adherence to unhealthy diets. In the present study, we evaluated the effects of mixed nuts supplementation on erectile and sexual function. Eighty-three healthy male aged 18–35 with erectile function assessment were included in this FERTINUTS study sub-analysis; a 14-week randomized, controlled, parallel feeding trial. Participants were allocated to (1) the usual Western-style diet enriched with 60 g/day of a mixture of nuts (nut group; *n* = 43), or (2) the usual Western-style diet avoiding nuts (control group; *n* = 40). At baseline and the end of the intervention, participants answered 15 questions contained in the validated International Index of Erectile Function (IIEF), and peripheral levels of nitric oxide (NO) and E-selectin were measured, as surrogated markers of erectile endothelial function. Anthropometrical characteristics, and seminogram and blood biochemical parameters did not differ between intervention groups at baseline. Compared to the control group, a significant increase in the orgasmic function (*p*-value = 0.037) and sexual desire (*p*-value = 0.040) was observed during the nut intervention. No significant differences in changes between groups were shown in peripheral concentrations of NO and E-selectin. Including nuts in a regular diet significantly improved auto-reported orgasmic function and sexual desire.

## 1. Introduction

National Institutes of Health (NIH) define erectile dysfunction (ED) as a persistent difficulty achieving and maintaining an erection sufficient for satisfactory sexual intercourse [1]. The prevalence of ED ranged from 2% in men younger than 40 years old, around 52% in men aged 40–70 years, and more than 85% in men with 80 years and older [2,3]. Although significant advances in the field were made, primary prevention research on this condition is very preliminary. Lifestyle risk factors for ED include smoking, excessive alcohol consumption, lack of physical activity, psychological stress, overweight or obesity, and adherence to unhealthy diets, among others [4,5,6,7].

Lifestyle factors may influence ED through the vascular and nervous system. Because adequate arterial supply is critical for erection, any disorder that impairs blood flow may be implicated in the etiology of erectile failure. However, a wide variety of psychological problems can influence the male erectile response because, in a vascular event initiated by neuronal action, it is maintained by a complex interplay between vascular, neurological events and other comorbidities [7,8]. Moreover, it is generally accepted that nitric oxide (NO) is the principal agent responsible for relaxation/erection of penile smooth muscle. 

Mediterranean diet and some components of the Mediterranean diet have been inversely related to erectile and sexual dysfunction [9] but also a better endothelial function [10]. This is the case of nuts that its consumption has consistently demonstrated beneficial effects on endothelial function [11]. In fact, in a recent study, it has demonstrated that pistachio consumption improves erectile function, probably because it contains (as other types of nuts [12]) several antioxidants, and arginine, a precursor of nitric oxide (NO), a powerful compound that increases vasodilatation [13]. 

To demonstrate that a dietary pattern or a food group can not only modulate the erectile function, but also the sperm function increasing the chances of fertility is of great interest mainly in developed countries where male infertility seems to have fallen drastically [14], and the psychological stress seems to be more prevalent [15].

Therefore, in order to elucidate the possible role of nut consumption in the primary prevention of ED, we explored using a randomized controlled trial (RCT), the effects of nuts supplementation on erectile function determined by the International Index of Erectile Function, but also the endothelial function by measuring peripheral concentrations of NO and E-selectin.

## 2. Materials and Methods

### 2.1. Study Design

The study design of the FERTINUTS trial has been reported previously [16]. The trial was registered in ISRCTN registry with identifier ISRCTN12857940. Briefly, FERTINUTS was a 14-week (wk) randomized, controlled, two-interventions parallel, clinical trial conducted in healthy males who reported a Western-style diet. The trial was conducted between 2015 and 2017, and included participants who were randomly assigned (1:1) to one of the following two interventions: (1) enriching the usual Western-style diet with 60 g/d of a mixture of raw walnuts, almonds, and hazelnuts (nut group); or (2) following the usual Western-style diet avoiding nuts (control group). The protocol was approved by the Institutional Review Board of the *Hospital Universitari Sant Joan de Reus* in October 2015. All the participants provided a written informed consent.

Eligible participants were healthy men aged 18–35 years old. The following exclusion criteria were applied: frequent consumption of nuts or a known history of allergy; use of plant sterol or fish oil supplements and multivitamins, vitamin E or other antioxidant supplements; history of reproductive disorders or vasectomy; current smokers; medications for chronic illness consumption; or use of illegal drugs. More detailed criteria for enrolment have been reported elsewhere [16]. 

The effect of the interventions on several cardiovascular risk factors and sperm parameters have been reported previously [16,17,18]. We report here the effect of the interventions on auto-reported erectile function parameters and the concentrations of peripheral endothelial biomarkers over 14-wk as a secondary outcome.

### 2.2. Anthropometric, Dietary, Blood Parameters, and Seminogram Measurements

Trained nurses, biologists, or dietitians directly recall all the general participants’ information and conduct anthropometric measurements. The initial assessment of individuals was conducted with a 15-item dietary screener modified from Martínez-González et al., 2012 [19] to verify the presence of a Western-style diet adherence. Participants in the nut group received at no cost, every month, pre-weighed packs for the consumption of 60 g of nuts per day (30 g of walnuts, 15 g of almonds, and 15 g of hazelnuts). Participants in both groups received detailed dietary instructions in order to increase the adherence to the assigned intervention.

At baseline, participants completed a general questionnaire with a medical history, reproductive history, use of medication, and a 143-item semi-quantitative validated food frequency questionnaire (FFQ) [20] in a face-to-face interview.

During the 14-wk follow-up (with four in-site visits), weight, height, and waist circumference were recorded using a high-precision electronic scale (TANITA TBF-300, Tanita). Blood pressure was measured at rest in duplicate with a 5 minutes interval between each measurement by using a semiautomatic oscillometer (Omron HEM-705CP, Netherlands). Furthermore, all the participants completed a specific questionnaire reporting any adverse effects related or not related to the intervention, and a 3-day dietary record (3DDR) in a face-to-face interview with an expert dietitian in order to measure the compliance with the dietary intervention. Energy and nutrient intake were calculated using Spanish food composition tables [21,22]. Adherence to the intervention was also assessed by counting the empty sachets of nuts returned in each visit.

At baseline and at the end of the intervention, blood samples in 12 h fasting conditions and semen samples after 3 days of sexual abstinence were collected. Fasting glucose, total cholesterol, HDL cholesterol, LDL cholesterol, VLDL cholesterol, triglycerides, insulin, C-reactive protein (CRP), and folate were determined (COBAS; Roche Diagnostics Ltd, UK) in blood. Semen volume and pH, sperm count and concentration, sperm motility, sperm viability, and sperm morphology were assessed in semen following the 2010 WHO criteria and the Björndahl checklist [23,24].

### 2.3. Erectile Function Questionnaire

The main outcome in the present analysis was the erectile function. To evaluate the influence of nuts on erectile function, participants answered to the 15 questions contained in the validated International Index of Erectile Function (IIEF) [25] at baseline and the end of the intervention. The IIEF-15 permits to detect treatment-related changes [25,26]. The questionnaire of IIEF-15 addresses the relevant domains of male sexual function: erectile function (EF: 0–6 severe dysfunction, 7–12 moderate dysfunction, 13–18 mild to moderate dysfunction, 19-24 mild dysfunction, 25–30 no dysfunction), orgasmic function (OF: 0–2 severe dysfunction, 3-4 moderate dysfunction, 5–6 mild to moderate dysfunction, 7–8 mild dysfunction, 9–10 no dysfunction), sexual desire (SD: 0–2 severe dysfunction, 3–4 moderate dysfunction, 5–6 mild to moderate dysfunction, 7–8 mild dysfunction, 9–10 no dysfunction), intercourse satisfaction (IS: 0–3 severe dysfunction, 4–6 moderate dysfunction, 7–9 mild to moderate dysfunction, 10–12 mild dysfunction, 13–15 no dysfunction), and overall satisfaction (OS: 0–2 severe dysfunction, 3–4 moderate dysfunction, 5–6 mild to moderate dysfunction, 7–8 mild dysfunction, 9–10 no dysfunction).

### 2.4. Surrogated Measures of Erectile Endothelial Function

At baseline and end of the trial the peripheral concentrations of endothelial function markers, NO and E-selectin, were measured by Enzyme-Linked Immunosorbent Assay (ELISA) procedures according to the manufacturer instructions. Briefly, the NO assay kit (ThermoFisher Scientific) determines nitric oxide composition through measurement of nitrate (NO^3^) and nitrite (NO^2^) levels, while the E-selectin assay kit (ThermoFisher Scientific) determines the soluble E-selectin. Samples were read at 540 nm absorbance in the case of NO assay, and 450 nm absorbance in the case of E-selectin assay (TECAN, Sunrise). A polynomial curve was used as the standard. Laboratory technicians were blinded to group assignments.

### 2.5. Statistical Analyses

The sample size for the FERTINUTS trial was calculated to detect significant differences in the viability after nut consumption based on the results of Robbins et al., 2012 [27]. However, taking into account EF changes reported in a previous similar trial [9], a total sample size of 54 (27 per arm) was estimated to provide sufficient statistical power (more than 80%) to assess the effects of nut supplementation on erectile function parameters assuming two-sided 95% confidence interval.

Kolmogorov–Smirnov and Levene’s test were used in order to check the normal distribution and homogeneity, respectively. The data are shown as means ± standard deviation (SD) for normally distributed variables, and median ± interquartile rank (IQR) for non-normal continuous variables. Non-parametric statistical Mann–Whitney for non-paired data and Wilcoxon tests for paired data were used to assess differences within each intervention. ANCOVA models were applied to assess differences in changes between intervention groups after adjusting for baseline values. Spearman correlation coefficients were used to calculate pair-wise correlations, and Benjamini-Hochberg false discovery rate (FDR) correction was used for multiple comparisons. Statistical analyses were conducted using per protocol approaches, including all randomized participants fulfilling all baseline and final measurements. *p*-values of <0.05 were considered statistically significant. Statistical analyses were carried out using the freely available R statistical computing environment v.2.14.2 (www.r-project.org) [28] and the additional package Deducer for R (http://www.deducer.org/) [29].

## 3. Results

In the FERTINUTS trial, we assessed 244 subjects for eligibility. Of these, 57 subjects declined to participate and 68 did not meet the inclusion criteria. Thus, 119 participants were included in the trial and were randomly assigned to one of the two intervention groups: 61 in the nuts group and 58 in the control group. A total 98 participants successfully completed the study, and finally, 83 participants were included in this secondary analysis (those subjects who fulfilled the International Index of Erectile Function questionnaire): 43 in the nuts supplemented group and 40 in the control group (Figure 1).

Baseline characteristics (age, weight, height, BMI, waist circumference, systolic and diastolic blood pressure, fasting glucose, serum total cholesterol, HDL cholesterol, LDL cholesterol, VLDL cholesterol, triglycerides, insulin, C-reactive protein and folate, and main sperm parameters) are detailed in Table 1. No significant differences were observed in these baseline parameters agreeing to the sequence of randomization. Participants in the two groups reported similar adherence to the Western-style diet at baseline according to the 15-item dietary screener.

Compliance with the intervention, as assessed by counting the empty sachets of nuts returned by the participants, was high (>95% of empty sachets returned). According to the 3DDR data, significant between-group differences in nut intake was shown through the study. This was associated with an increase in the intake total fat (*p*-value < 0.001), MUFA (*p*-value<0.001), PUFA (*p*-value < 0.001), magnesium (*p*-value < 0.001), vitamin E (*p*-value = 0.014), omega-3 fatty acids (*p*-value < 0.001), α-Linolenic acid (ALA) (*p*-value < 0.001), and omega-6 fatty acids (*p*-value = 0.016) in the nut-supplemented group. The intake of energy (*p*-value = 0.029) and fiber (*p*-value = 0.002) experienced a smaller decrease in the nut-supplemented group compared with the control group (Table 2).

No significant between-group differences were observed in erectile function parameters at baseline. However, compared to the control group, a significant increase in the orgasmic function (OF; *p*-value = 0.037) and sexual desire (SD; *p*-value = 0.040) was observed in the nut-supplemented group during the intervention. No significant between-group differences in changes during the intervention were found in erectile function (EF; *p*-value = 0.192), intercourse satisfaction (IS; *p*-value = 0.473), and overall satisfaction (OS; *p*-value = 0.333) (Figure 2). No significant correlations were found between changes in ED parameters and changes in biochemical parameters during the intervention.

Moreover, no significant differences in changes between intervention groups were shown in peripheral concentrations of NO (*p*-value = 0.737) (Figure 3A) or E-selectin (*p*-value = 0.347) (Figure 3B).

## 4. Discussion

Herein we report that adding 60 g/d of mixed raw nuts to a Western-style diet for 14-wk improved the auto-reported orgasmic function and sexual desire parameters in a group of healthy reproductive-aged participants compared with an age-matched control group. In the present study, none of the possible mechanisms explored (NO and E-selectin as surrogated markers of endothelial function) seem to explain the beneficial effects observed on orgasmic function and sexual desire.

Interestingly, our findings in healthy young males are pretty consistent with the only previous clinical study reporting an increase of all the five IIEF-15 domains after 100 g/day pistachio consumption for 3 weeks [13], although this study was conducted in patients with erectile dysfunction at baseline. Therefore, our study extends the findings to a healthy population without erectile dysfunction supplemented with a mixture of nuts like hazelnuts, almonds, and walnuts.

Nuts are nutrient-dense foods with a special nutrient content, a key component of several healthy dietary patterns and recommendations, and its consumption is associated with improvements in some cardiovascular disease risk factors [30,31,32]. Specifically, hazelnuts, almonds and walnuts contain high amounts of vegetable protein and fat (mainly unsaturated fatty acids), are dense in antioxidants and vitamins (e.g., folic acid, niacin, tocopherols, and vitamin B6, among others) and some minerals (e.g., calcium, magnesium, phosphorous and potassium), and also rich = in dietary fiber and many other bioactive constituents (e.g., phytosterols and phenolic compounds) (Appendix A [33]).

In addition, nuts had a relatively high amount of the nonessential amino acid arginine, a precursor of NO, that is a potent vasoactive neurovascular, nonadrenergic, noncholinergic (NANC) neurotransmitter that plays an important role in erectile action through the corpora cavernosa [34]. The results from our study do not demonstrate that Arginine-NO pathway act as the unique player modulating erectile function. However, we cannot discard a lack of statistical power to demonstrate differences between intervention groups in relation to these subrogated markers of endothelial function, because the sample size of the present sub-analysis was estimated using the IIEF as the main outcome.

Another promising serum biomarker for erectile function is serum E-selectin [35]. In that case, E-selectin, because it is an endothelial dysfunction marker, seems more useful in patients diagnosed with diabetes mellitus [36]. E-selectin is a cell adhesion molecule activated by cytokines that plays an important role in inflammation. Because consuming between 60 and 90 g of nuts has proven effective improving inflammation [37] it could have been reasonable to detect some differences in this marker due to the nut’s supplementation. Nonetheless, our study does also not confirm any effect on this endothelial marker. This lack of effect could be explained not only because of a lack of power but also because our participants were healthy and therefore without having type 2 diabetes.

Because we detected an improvement in the auto-reported orgasmic function and sexual desire parameters, maybe other mechanisms beyond those mentioned above may be implicated. It is interesting to mention that erectile (dys)function and atherogenesis share common pathways [38]. For that reason, several antioxidants (e.g., polyphenols) and vitamins, that are present in nuts, have been suggested to be effective treatments for ED and at the same time are beneficial for the cardiovascular system [38]. Previous studies reported that chronic consumption of nuts has proven effective for lowering LDL cholesterol [32] and improving glucose metabolism [39], among other cardiovascular risk factors, decreasing the incidence of major cardiovascular events [17]. Therefore, we strongly believe in the necessity to develop similar trials with participants at high cardiovascular risk and erectile dysfunction to accurately establish an effect of nut consumption on erectile function and cardiovascular risk.

Our study has several strengths. This is the largest and unique RCT to date analyzing the effect of nut supplementation on erectile and sexual function in healthy participants. Moreover, the present analysis has theoretically enough statistical power to detect effects on erectile function measured by a validated International Index of Erectile Function. Although several questionnaires have been developed to objectively evaluate EDs, the validated International Index of Erectile Function (IIEF) questionnaire is considered a valuable tool for defining the area of sexual dysfunction that may be incorporated as part of the clinical history to document the degree of dysfunction and gauge response to therapy [25]. Moreover, having detailed information on medical and reproductive history allowed us to reduce bias by excluding participants with chronic and reproductive diseases that may influence diet, seminogram or erectile function. The main strength of the present study is the design because RCTs represent the cornerstone of evidence-based medicine. 

However, the following limitations need to be highlighted. The present study is based on a secondary outcome analysis of the FERTINUTS trial. Second, our study was conducted in apparently healthy and fertile participants limiting the extrapolation of the results to other populations, for example with erectile dysfunction, the inclusion of a group of subjects suffering from erectile dysfunction could help us to extend the results obtained. Third, reproductive hormonal (e.g., testosterone, prolactin, FSH, estradiol) values, which could affect erectile and sexual function are not reported. Finally, our study did not provide enough evidence to support the main mechanism for these improvements, however, an absence of evidence does not mean evidence of no effect [40]. Only equivalence trials are properly suited to demonstrate the equality of effects. For that reason, other RCT focused on markers of erectile endothelial function as possible mechanisms of the effect as main outcomes, are warranted in the future to increase the scientific evidence in the field.

## 5. Conclusions

In conclusion, our study suggests that compliance with a healthy diet supplemented with mixed nuts may help to improve erectile and sexual desire. Large studies are warranted in the future to confirm these results and elucidate possible mechanisms implicated on these benefits.

## Figures and Tables

**Figure 1 nutrients-11-01372-f001:**
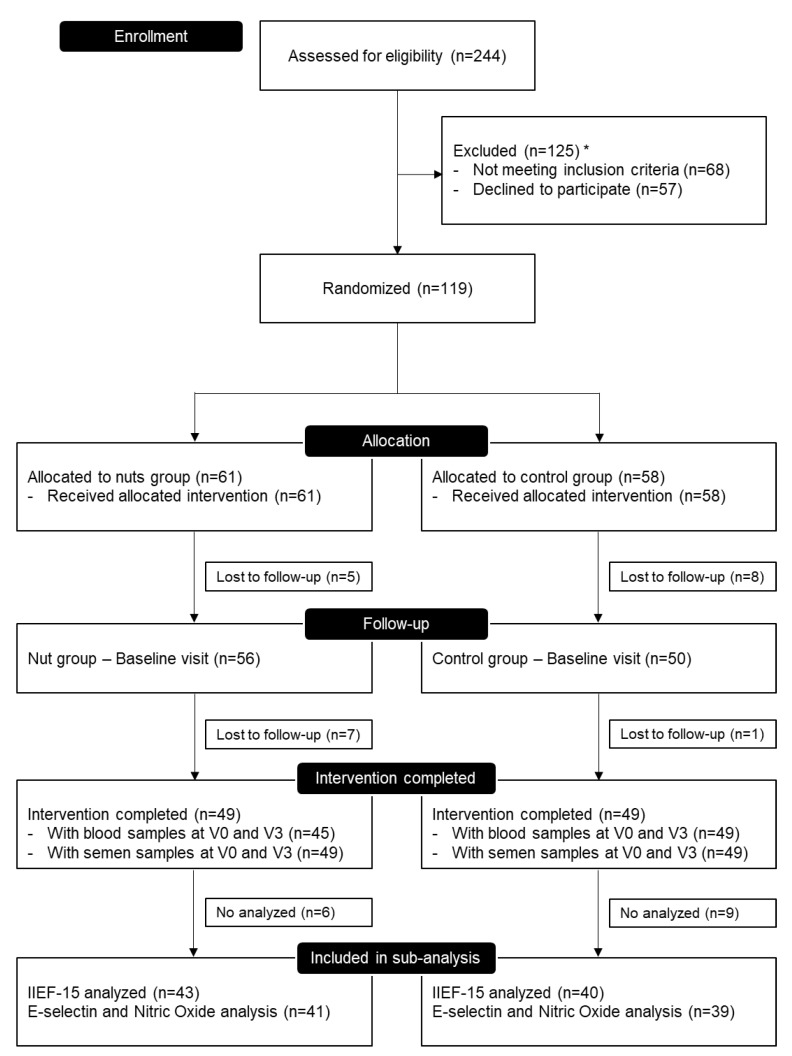
Flow diagram of the FERTINUTS sub-analysis. * 68 participants did not meet the inclusion criteria (ascribed to a non-Western style diet, *n* = 37; smoking, *n* = 21; or other minor reasons, *n* = 10) and 57 subjects declined to participate (lack of interest, *n* = 34; impossible to contact with them, *n* = 18; and for non-economic compensation, *n* = 5).

**Figure 2 nutrients-11-01372-f002:**
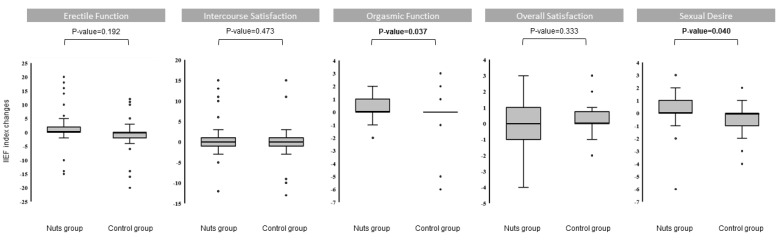
Box plot of the differences between the control group and the nut-supplemented group in the erectile function parameters (IIEF-15). ANCOVA models were used to assess differences between intervention groups. Changes were adjusted for baseline values. A horizontal line in the boxplot illustrates the median value. The upper and lower bars indicate the third and first percentiles, respectively. Outliers are plotted as individual circles. Abbreviations: IIEF: International Index of Erectile Function.

**Figure 3 nutrients-11-01372-f003:**
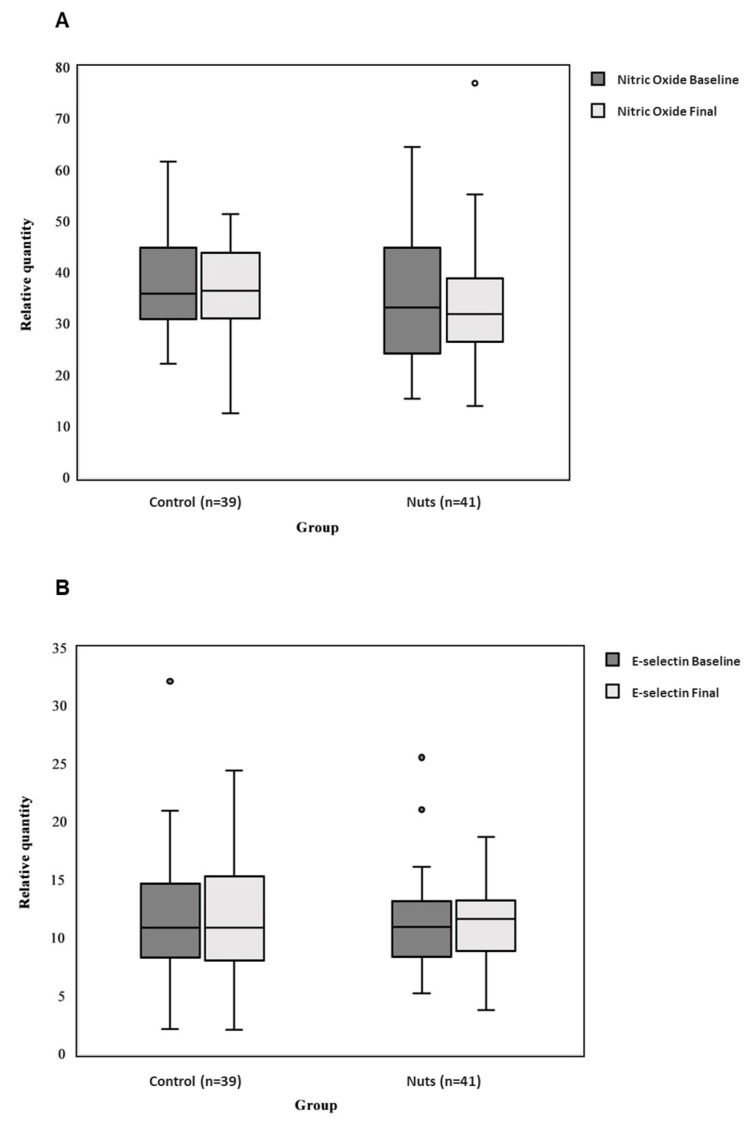
Box plot of the differences between the control group and the nut group in the concentrations of nitric oxide (**A**) and E-selectin (**B**) erectile function markers. ANCOVA models were used to assess differences between intervention groups. Changes were adjusted for baseline values. A horizontal line in the boxplot illustrates the median value. The upper and lower bars indicate the third and first percentiles, respectively. Outliers are plotted as individual circles. In this analysis 3 participants (2 participants in the nut group and 1 in the control group) had missing values for NO or E-selectin (see Figure 1).

**Table 1 nutrients-11-01372-t001:** Baseline characteristics of the study population (general characteristics, blood, and semen parameters).

Variable	Nuts Group (*n* = 43)	Control Group (*n* = 40)	*p*-Value
General characteristics; mean (SD)			
Age (years)	24.05 (4.82)	25.83 (4.58)	0.066
Weight (kg)	73.44 (10.16)	76.82 (12.04)	0.270
Height (cm)	176.46 (6.21)	177.73 (6.66)	0.409
BMI (kg/m^2^)	23.53 (2.59)	24.33 (3.64)	0.379
Waist circumference (cm)	79.98 (7.41)	83.26 (8.98)	0.067
Systolic blood pressure (mmHg)	129.38 (11.37)	126.50 (11.79)	0.425
Diastolic blood pressure (mmHg)	73.34 (7.90)	70.90 (8.49)	0.174
Score for adherence to Western-style diet ^a^	8.11 (2.16)	8.80 (2.26)	0.182
Blood parameters; median (IQR)			
Fasting plasma glucose (mg/dl)	87.0 (82.0, 93.5)	85.5 (81.5, 91.0)	0.384
Total cholesterol (mg/dl)	167.0 (149.5, 188.5)	173.5 (150.0, 196.0)	0.374
HDL-c (mg/dl)	58.0 (49.5, 66.5)	55.5 (50.0, 67.0)	0.736
LDL-c (mg/dl)	87.0 (74.5, 105.5)	98.0 (77.8, 119.3)	0.202
VLDL-c (mg/dl)	13.0 (12.0, 19.0)	13.0 (9.0, 17.3)	0.240
Triglycerides (mg/dl)	66.0 (59.0, 95.5)	64.0 (47.0, 85.0)	0.258
Fasting plasma insulin (mcUl/ml)	5.40 (2.60, 8.60)	5.20 (2.75, 6.80)	0.654
C-Reactive protein (mg/dl)	0.20 (0.20, 0.20)	0.20 (0.13, 0.20)	0.144
Folate (ng/ml)	6.30 (4.80, 8.70)	6.30 (4.88, 7.80)	0.616
Semen characteristics; median (IQR)			
pH	8.0 (8.0, 8.5)	8.0 (8.0, 8.5)	0.940
Volume (mL)	3.50 (1.95, 4.55)	3.40 (2.50, 5.13)	0.179
Total spermatozoa (×10^6^)	75.20 (28.10, 104.50)	72.05 (40.90, 125.50)	0.497
Spermatozoa concentration (×10^6^)	25.20 (14.50, 41.80)	19.80 (9.70, 37.95)	0.402
Viability (%)	78.68 (70.26, 82.52)	80.21 (73.66, 85.87)	0.257
Total motility (progressive and non-progressive motility) (%)	64.66 (45.91, 71.34)	70.11 (62.73, 78.56)	0.097
Progressive motility (%)	44.67 (28.27, 53.97)	49.72 (35.39, 61.67)	0.086
Non-progressive motility (%)	13.26 (9.79, 16.15)	11.64 (7.58, 14.31)	0.103
Immotile spermatozoa (%)	35.33 (28.58, 52.06)	29.89 (21.44, 37.27)	0.094
Normal forms (%)	6.33 (4.91, 8.17)	6.27 (5.23, 7.57)	0.935
Abnormal head (%)	52.59 (43.06, 66.28)	55.07 (41.22, 67.81)	0.771
Abnormal midpiece (%)	10.71 (8.63, 15.05)	12.71 (8.46, 14.23)	0.705
Abnormal principal piece (%)	12.94 (5.44, 29.66)	9.46 (4.53, 25.27)	0.529
Combined abnormality (%)	8.40 (6.49, 12.88)	7.85 (6.71, 14.07)	0.985

Data are given as mean and standard deviation (SD) or medians and Interquartile ranges (IQRs). All the analyses were assessed by non-parametric tests (the Mann–Whitney for non-paired data) for normality distribution reasons. Equivalences: 1 mg/dl plasma glucose = 18.018 mmol/l, 1 mg/dl total cholesterol= 38.610 mmol/l. ^a^ The score for adherence to the Western-style diet is based on a 15-item dietary screener (a score of zero indicates minimum adherence, a score of 15 indicates maximum adherence). Abbreviations: BMI, body mass index; HDL, high-density lipoprotein; IQR, interquartile rank; LDL, low-density lipoprotein; SD, standard deviation; VLDL, very-low-density lipoprotein.

**Table 2 nutrients-11-01372-t002:** Nutrient intake at baseline and changes by intervention group.

Variables	Nut Group (*n* = 43)		Control Group (*n* = 40)		Treatment Effect
	Baseline	Changes	Baseline	Changes	*p*-Value
Energy					
Energy intake (kcal/d)	2699.13 (994.43)	−145.19 (31.74)	2359.57 (565.72)	−215.22 (18.18)	**0.029**
Macronutrients					
Proteins (g/d)	112.20 (39.37)	−7.68 (1.44)	103.13 (22.51)	−10.45 (0.34)	0.065
Carbohydrates (g/d)	305.24 (134.16)	−50.48 (4.83)	254.34 (63.04)	−11.21 (0.05)	0.333
Simple carbohydrates (g/d)	112.70 (79.80)	−13.47 (3.33)	92.54 (28.75)	−6.31 (0.06)	0.696
Complex carbohydrates (g/d)	176.44 (68.26)	−34.36 (0.84)	145.29 (46.70)	−12.09 (1.30)	0.604
Total fat (g/d)	106.89 (40.75)	12.19 (0.66)	97.98 (30.47)	−11.95 (0.92)	**<0.001**
MUFA (g/d)	42.87 (18.36)	5.89 (0.32)	37.97 (15.82)	−6.14 (0.48)	**<0.001**
SFA (g/d)	33.33 (17.87)	−3.19 (0.21)	31.17 (10.64)	−3.67 (0.04)	0.589
PUFA (g/d)	13.51 (6.41)	11.03 (0.27)	11.37 (4.50)	−2.60 (0.23)	**<0.001**
Cholesterol (mg/d)	426.64 (253.96)	−70.48 (8.95)	386.09 (131.85)	−56.08 (2.37)	0.745
Fiber (g/d)	24.69 (11.21)	−0.77 (0.39)	19.84 (9.45)	−2.39 (0.41)	**0.002**
Alcohol (g/d)	9.43 (14.01)	−2.79 (0.73)	7.91 (13.14)	−2.25 (0.89)	0.693
Glycemic load	172.69 (75.43)	−31.38 (2.55)	141.20 (37.69)	−5.52 (0.32)	0.384
Glycemic index	56.16 (4.50)	−1.48 (0.18)	55.13 (5.45)	0.29 (0.08)	0.346
Micronutrients					
Sodium (mg/d)	3452.88 (1455.36)	−407.37 (26.13)	3178.08 (958.75)	−174.95 (23.37)	0.743
Potassium (mg/d)	3639.91 (1624.75)	−217.88 (58.80)	3126.17 (968.16)	−140.77 (21.62)	0.289
Linoleic (g/d)	26.08 (26.64)	4.79 (1.35)	31.40 (31.79)	−4.66 (1.24)	0.311
Magnesium (mg/d)	407.62 (199.96)	32.17 (7.96)	329.53 (108.61)	−16.92 (0.93)	**<0.001**
Calcium (mg/d)	987.45 (514.83)	−98.27 (2.79)	898.70 (324.53)	−51.64 (13.16)	0.843
Iron (mg/d)	20.27 (14.55)	−2.04 (0.60)	18.35 (13.33)	−0.11 (0.47)	0.864
Selenium (mg/d)	222.04 (162.30)	−48.94 (13.01)	209.25 (140.86)	−46.06 (11.51)	0.563
β-carotene equivalents (µg/d)	4711.11 (5096.91)	−1008.75 (348.71)	3913.42 (4116.64)	22.34 (72.89)	0.642
Retinol (µg/d)	592.05 (1409.67)	−230.55 (169.57)	402.76 (166.28)	141.07 (141.29)	0.254
Vitamin D (µg/d)	7.37 (13.90)	0.55 (0.31)	5.05 (5.65)	4.50 (3.67)	0.606
Vitamin E (µg/d)	13.18 (5.12)	6.10 (1.20)	15.19 (18.38)	−3.03 (0.93)	**0.014**
Vitamin K (µg/d)	130.94 (163.71)	−10.50 (0.30)	115.38 (111.22)	−32.51 (6.99)	0.213
Omega-3 (g/d)	1.72 (0.89)	1.53 (0.02)	1.68 (0.73)	−0.10 (0.02)	**<0.001**
ALA (g/d)	1.10 (0.61)	1.55 (0.02)	1.00 (0.39)	−0.22 (0.02)	**<0.001**
EPA (g/d)	0.51 (2.74)	0.02 (0.00)	1.03 (3.96)	0.65 (0.12)	0.236
DHA (g/d)	0.15 (0.19)	0.02 (0.01)	0.21 (0.31)	0.10 (0.03)	0.161
Omega-6 (g/d)	18.03 (14.80)	10.27 (0.08)	24.64 (23.36)	−3.76 (0.94)	**0.016**

Data are given as means and SD for baseline values, and mean and SE for changes. In bold the significant values. ANCOVA models were used to assess differences between intervention groups. Changes in variables were adjusted for baseline values of each variable. Abbreviations: ALA: α-Linolenic acid, DHA: docosahexaenoic acid, EPA: eicosapentaenoic acid, MUFA: monounsaturated fatty acids, PUFA: polyunsaturated fatty acids, SFA: saturated fatty acids.

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
