# Peer review of "Effect of Nut Consumption on Erectile and Sexual Function in Healthy Males: A Secondary Outcome Analysis of the FERTINUTS Randomized Controlled Trial"

_nutrients, 2019, doi:10.3390/nu11061372_

Reviewer 1 Report

This is a very interesting randomized control trial that aims at determining the effects of diet (nuts) on erectile function. The study involved a significant number of young individuals, and found that the diet supplemented with nuts increased the orgasmic function and sexual desire. In contrast, no significant differences between conventional spermiogram parameters were found.

The Manuscript is interesting, falls into the Journal’s scope, the techniques and stats are sound, and data are adequately discussed. I have no major concerns on the scientific approach or content of the paper. I would just suggest using ‘viability’ rather than ‘vitality’ for the evaluation of sperm membrane integrity.

Author Response

Reviewer #1:

This is a very interesting randomized control trial that aims at determining the effects of diet (nuts) on erectile function. The study involved a significant number of young individuals, and found that the diet supplemented with nuts increased the orgasmic function and sexual desire. In contrast, no significant differences between conventional spermiogram parameters were found.

The Manuscript is interesting, falls into the Journal’s scope, the techniques and stats are sound, and data are adequately discussed. I have no major concerns on the scientific approach or content of the paper.

We sincerely thank Reviewer #1 for the global appreciation of our investigation, as well as for all the valuable comments and suggestions provided in the following lines, which have greatly improved the first version of the manuscript. We have addressed all them in each of the following points, as well as in the manuscript, when required. Please find below the itemized responses to all Reviewer #1’s comments.

I would just suggest using ‘viability’ rather than ‘vitality’ for the evaluation of sperm membrane integrity.

We thank the reviewer for this comment. Changed as suggested through all the manuscript.

Reviewer 2 Report

In the present study, the Authors evaluated the effects of mixed nuts supplementation on erectile and sexual function in a randomized, controlled, parallel feeding trial, using a validated International Index of Erectile Function (IIEF), at baseline and the end of the intervention.

In parallel, peripheral levels of Nitric Oxide (NO) and E-selectin were also measured, as surrogated markers of erectile endothelial function.

A significant increase in the orgasmic function and sexual desire was observed during the nut intervention, while no significant differences in changes between groups were shown in peripheral concentrations of NO and E-selectin.

The manuscript is interesting and well written; the statistical analysis is well conducted.

The tables and figures are clear (in figure 2 the box plot related to the orgasmic function is missing); and the Supplemental Table 1 (average nutrient composition of studied nuts) is very useful.

However, there are limits:

- anthropometrical characteristics, seminal and blood biochemical parameters are indicated in the studied groups, but the hormonal values are not reported (above all, testosterone and prolactin, which greatly affect sexuality)

- the manuscript could have a more significant value if a group of subjects suffering from erectile dysfunction is included in the study.

Author Response

Reviewer #2:

In the present study, the Authors evaluated the effects of mixed nuts supplementation on erectile and sexual function in a randomized, controlled, parallel feeding trial, using a validated International Index of Erectile Function (IIEF), at baseline and the end of the intervention.

In parallel, peripheral levels of Nitric Oxide (NO) and E-selectin were also measured, as surrogated markers of erectile endothelial function.

A significant increase in the orgasmic function and sexual desire was observed during the nut intervention, while no significant differences in changes between groups were shown in peripheral concentrations of NO and E-selectin.

The manuscript is interesting and well written; the statistical analysis is well conducted.

We sincerely thank Reviewer #2 for the global appreciation of our study, as well as for all the valuable comments and suggestions provided in the following lines, which have greatly improved the first version of the manuscript. We have addressed all them in each of the following points, as well as in the manuscript, when required. Please find below the itemized responses to the comments.

The tables and figures are clear (in figure 2 the box plot related to the orgasmic function is missing); and the Supplemental Table 1 (average nutrient composition of studied nuts) is very useful.

We uploaded again the Figure 2 to fix this issue.

However, there are limits:

- anthropometrical characteristics, seminal and blood biochemical parameters are indicated in the studied groups, but the hormonal values are not reported (above all, testosterone and prolactin, which greatly affect sexuality)

We completely agree with this comment. We have included this point in the discussion section as a limitation of our study (page 13, lines 286-288).

- the manuscript could have a more significant value if a group of subjects suffering from erectile dysfunction is included in the study.

We thank the reviewer for this comment however, the population comes from a project called FERTINUTS with a primary outcome described in detail in the following article: “Salas-Huetos, A.; Moraleda, R.; Giardina, S.; Anton, E.; Blanco, J.; Salas-Salvadó, J.; Bulló, M. Effect of nut consumption on semen quality and functionality in healthy men consuming a Western-style diet: a randomized controlled trial. Am. J. Clin. Nutr. 2018, 108, 953–962.”. Because the population is well defined previously the group of subjects suffering erectile dysfunction cannot be included in this secondary analysis of the same FERTINUTS population. However, we have expanded a little more this issue in the Discussion limitation part (page 13, lines 285-286).

Reviewer 3 Report

The manuscript entitled Effect of nut consumption on erectile and sexual function in healthy males: a secondary outcome analysis of the FERTINUTS randomized controlled trial by Albert Salas-Huetos et al. attempt to present the effect of nut supplementation on erectile and sexual function. The experiment and its results are well designed, properly written, very interesting, and present high research skills of the Authors and their team.

There are only a few minor editing issues that will make your manuscrypt easier to read. 

Table 1. Widening the "group of nuts" column makes it easier to read the results

Fig 2. Names of groups of erectile function parameters are written in a very small font - it could be a bit bigger.

Fig 2 and 3. The figures footnotes are written in different font sizes.

Author Response

Reviewer #3:

The manuscript entitled Effect of nut consumption on erectile and sexual function in healthy males: a secondary outcome analysis of the FERTINUTS randomized controlled trial by Albert Salas-Huetos et al. attempt to present the effect of nut supplementation on erectile and sexual function. The experiment and its results are well designed, properly written, very interesting, and present high research skills of the Authors and their team.

We sincerely thank Reviewer #3 for the global appreciation of our study, as well as for all the valuable comments and suggestions provided in the following lines, which have greatly improved the first version of the manuscript. We have addressed all them in each of the following points, as well as in the manuscript, when required. Please find below the itemized responses to the comments.

There are only a few minor editing issues that will make your manuscript easier to read.

Table 1. Widening the "group of nuts" column makes it easier to read the results.

We have changed the “nuts group” and “control group” columns widening to facilitate the read of the results in Table 1.

Fig 2. Names of groups of erectile function parameters are written in a very small font - it could be a bit bigger.

We thank the reviewer for this comment. Changed as suggested. Font size 12.

Fig 2 and 3. The figures footnotes are written in different font sizes.

We have changed the footnotes font sizes in Figures 2 and 3.

Round  2

Reviewer 2 Report

The authors have described in more detail the limits of this research.